# Research on Dynamic Response of a Single-Tower Cable-Stayed Bridge with Successive Cable Breaks Based on a 3D Index

Yongjian Chen [1], Song Wang [1], Kai Huang [1,*], Jiwei Zhong [2] and Hui Cheng [2]



1   College of Civil Engineering, Fuzhou University, Fuzhou 350108, China; chenyongjian@fzu.edu.cn (Y.C.);
    ws2020@foxmail.com (S.W.)
2   Railway Bridge Science Research Institute, Ltd., Wuhan 430034, China; jw_zhong@sohu.com (J.Z.);
    sockhui@163.com (H.C.)
*   Correspondence: huangkai@fzu.edu.cn

**Abstract:** Aiming at the problem of cable breakage of an asymmetric single-tower cable-stayed bridge by taking a single-tower cable-stayed bridge that has been in operation for more than 20 years as an engineering example, a spatial finite element model that reflected the actual stress state of the structure was established and a method for solving the time point of cable breakage was proposed. The results show that the strength of the main girder, pylon and cable could meet the stress requirements of the vibration caused by successive cable breaks, and the maximum dynamic displacement of the main girder did not exceed the stiffness limit. Although the breakage of the stay cable caused the stress in the adjacent position and the corresponding position on the opposite side of the tower to increase sharply, the stress fluctuation in other positions was small. The fluctuation of the dynamic stress of the cable tower was relatively small, but the displacement of the tower top changed greatly. The dynamic response of the cables breaking in succession was less than that of simultaneous breaking, and the structural vibration caused by cable breaking was not as severe as that caused by simultaneous breaking.

**Keywords:** single tower; cable-stayed bridge; successive cable breaking; 3D index; dynamic response

## 1. Introduction

A cable-stayed bridge is a composite structural bridge, which is composed of cables in tension, towers in compression, and beams in both bending and compression. Because of its good mechanical performance and strong spanning ability, the structure has been widely used and developed. However, there have been many cable breakage accidents on cable-stayed bridges in recent decades. For example, the main girder of Maracaibo Bridge partially collapsed due to broken cables. In China, Haiyin Bridge was forced to stop for maintenance due to broken cables, and the main girder of Chishi Bridge cracked in a large area due to continuous broken cables. From this point of view, the structural safety problem caused by cable fracture must be highly valued.

For cable-stayed bridges with broken cables, Zhao Xiang [1] designed a 1/60th scale model of Runyang Bridge to study the influence of cable breakage on the static performance of a cable-stayed bridge. Mozos [2] obtained the total fracture time through cable tension tests with three different loading rates and then studied the influence of cable fracture duration on structural dynamic response. Ma Yafei et al. [3] studied the influence of cable breakage on the static performance of a cable-stayed bridge through a segmental scale model and found that a medium-length breakage has a greater influence on a cable-stayed bridge than short cables and long cables. Michaltsos G.T. et al. [4,5] derived the two-dimensional and three-dimensional analytical algorithm equations of the cable breakage of cable-stayed bridges and solved them to obtain the displacement time–history function of the main girder after cable breakage. Chin-Sheng Kao and Chang-Huan Kou [6] studied the influence of a single cable fracture on the stress of a cable-stayed bridge structure and

found that the fracture of a single cable will significantly increase the cable force of adjacent cables. The fracture of a long cable will cause a large horizontal displacement of the cable tower and a large deflection in the middle of the main girder. Seungjun Kim and Young Jong Kang [7] used the finite element model to study the mechanical performance of a twin-tower steel cable-stayed bridge after the cable was broken. The results show that the failure of a single cable will significantly change the static performance and ultimate bearing capacity of the structure.

Regarding the dynamic analysis of cable breakage of cable-stayed bridges, Ruiz-Terana and Apariciob [8] studied the dynamic response of cable-stayed bridges under different cable-breaking conditions and pointed out that after 40% of the stay cables were broken, the design load of the structure did not reach the ultimate bearing capacity. Mozos et al. [9,10] studied the influence of different cable breaks on the mechanical properties of cable-stayed bridges and found that it is inaccurate to analyze the mechanical properties of main beams and cable towers after single cable breaks by using the pseudo-dynamic method with a coefficient of 2.0. Cai et al. [11] studied the cable failure of a cable-stayed bridge and found that the failure of a single cable will not cause the continuous collapse of a cable-stayed bridge structure, and the closer the failed cable is to the tower, the less likely it is to be broken continuously. Yukari Aoki et al. [12] studied the influence of the position, duration, number and damping ratio of broken cables on the continuous collapse resistance of cable-stayed bridges. Zhou and Chen [13,14] put forward a new method to simulate the sudden cable break of a cable-stayed bridge and studied the effects of random vehicle load, vehicle-bridge coupling vibration and cable break vibration on the dynamic response of a cable-stayed bridge structure. Zheng Xiaobo et al. [15] studied the bearing capacity decline of a cable-stayed bridge caused by cable failure and analyzed the dynamic response of a cable-stayed bridge in terms of the difference in cable-breaking time. Zhang Yu et al. [16] studied the dynamic response of a multi-tower cable-stayed bridge when several cables are broken. The results show that static cable breakage will only affect the internal force and cable force of the structure near the broken cable position, and local cable breakage will not cause the collapse of the whole bridge structure. Wang Tao et al. [17] used an implicit algorithm to study the dynamic response of a cable-stayed bridge under the combined action of wind, train and broken cable vibrations. The research shows that a single broken cable will not affect the train traffic.

Due to the complexity of the model test, the diversity of cable breakage factors and the high test cost, most scholars use finite element simulations to study the cable breakage of a cable-stayed bridge. However, scholars mostly study the twin towers and multi-tower cable-stayed bridges with double cable planes but seldom study the cable breakage of single-tower concrete beam cable-stayed bridges with an asymmetric single cable plane, and the dynamic response of cable breakage of single-tower cable-stayed bridges with long operation needs to be studied. Thus, a three-dimensional finite element of a single-tower cable-stayed bridge was established, and a method for solving the time point of cable failure was proposed. The cable that had the greatest influence on structural deformation was selected as the failure object, and the influence of cable failure on the dynamic response of a single-tower cable-stayed bridge was studied using a semi-dynamic method.

## 2. Evaluation Index and Analysis Method of Broken Cable

### 2.1. Evaluation Index

The research shows that most scholars study the cable fracture of a cable-stayed bridge by analyzing the internal force response of the structure. However, the actual internal force of a cable-stayed bridge in service is not easy to obtain, and thus, the structural internal force is not suitable for analyzing the stress state of a broken cable of a cable-stayed bridge in service. The component stress of a cable-stayed bridge is more suitable. Because the two ends of the cable are anchored on the main beam and the cable tower and the initial stress is applied, the main beam will be subjected to a strong axial force and become a

compression-bending member. Therefore, Formula (1) was adopted to calculate the stress of the main girder and pylon:

$$\sigma = \pm \frac{N}{A} + \frac{My}{I} \tag{1}$$

where N—axial force of the main girder and cable tower, M—bending moment of the main girder and cable tower, A—cross-sectional area, I—moment of inertia of the section, y—the distance from the calculated position to the neutral axis.

In a cable-stayed bridge structure, the sudden breaking of the cable means that the cable directly exits the structure. At the same time, it also causes structural vibration, and the stress and deformation of the structure reach the maximum response value in an instant. Then, the response value fluctuates within a certain range, finally tends to be in equilibrium under the damping effect and the remaining cables are forced to share the stress transferred by the broken cables. In order to study the influence of cable breakage on structural stress and deformation, the following three indexes (3D indexes) were proposed to analyze the dynamic response of an asymmetric single-tower cable-stayed bridge after a cable breakage. In the formulas, $S_0$ is the initial response value of structural vibration, $S_{dyn}$ is the maximum response value in the vibration process and $S_s$ is the stress value when the vibration tends to be static.

In order to quantify the maximum amplitude of structural stress and deformation during structural vibration, the dynamic influence coefficient (DIC) of a cable-stayed bridge was defined, and its value was greater than 1, which indicates that cable breakage is unfavorable to structural stress.

$$DIC = \left| \frac{S_{dyn}}{S_0} \right| \tag{2}$$

To study the influence of structural vibration caused by a cable breaking on cable stress, the definition of dynamic amplification factor by PTI [18] was introduced:

$$DAF_1 = \left| \frac{S_{dyn} - S_0}{S_s - S_0} \right| \tag{3}$$

To quantify the influence of structural vibration caused by the sudden breaking of cables on the stress and deformation of components other than cables and consider the static effect, the second dynamic amplification factor of cable-stayed bridges was introduced [19], and the formula is as follows:

$$DAF_2 = \left| \frac{S_{dyn}}{S_S} \right| \tag{4}$$

In order to quantify the influence of sudden cable breakage on the bearing capacity of cable-stayed bridge members, the definition of demand-capacity ratio is introduced, and it is considered that DCR > 1 members were destroyed due to material yield, and the calculation formula is as follows:

$$DCR = \left| \frac{\sigma_{max}}{\sigma_y} \right| \tag{5}$$

where $\sigma_{max} = S_{dyn}$, which is the maximum response value of structural stress. $\sigma_y$ is the yield strength, which is the strength standard value of the material. In this paper, the tensile strength standard value of cable steel wire and the compressive strength standard value of concrete were selected [20].

### 2.2. Analytical Method

The main methods used to study the sudden cable breakage of cable-stayed bridges are the pseudo-dynamic method, full dynamic method and semi-dynamic method. The pseudo-dynamic method [9,10] refers to the direct removal of the cable and the application of an equivalent load at its anchorage, which is twice the static cable force of the cable, and the static analysis is carried out on this basis. The full dynamic method [21] refers to



removing the cables that need to be broken, simulating the original structural state with the actual cable force, and then applying a pair of concentrated forces that change with time in the opposite direction at the cable anchorage position of the main beam and the cable tower. The whole analysis process is under dynamic conditions. The semi-dynamic method [21] refers to using the "life and death element" of ANSYS to simulate the sudden breaking of cables. The simulation process is a transient dynamic analysis starting with a static load step.

The research shows that the amplification factor of 2.0 in the pseudo-dynamic method cannot accurately describe the dynamic response of the structure [9,10]. Compared with the full dynamic method, the semi-dynamic method does not need to calculate the initial cable force separately or define the cable force change after the cable is broken. The semi-dynamic method can simulate the extreme situation of a cable breaking suddenly only by using "EKILL" to remove the cable unit at a certain moment. Therefore, a method was proposed to study the instantaneous cable breakage of a single-tower cable-stayed bridge, and the specific calculation process is as follows:

(1) Enter ANSYS and input the finite element model.
(2) Set an appropriate static analysis step size. The static analysis of the structure is carried out to obtain the initial stress and deformation of the structural cable before breaking.
(3) Set the step size of the dynamic analysis and perform the dynamic calculation. First, the cable element is broken in an instant, and the appropriate analysis step and integration time are specified. Second, in order to obtain the maximum dynamic response of the structure, the analysis sub-step is extended. Finally, the analysis step is extended again, and the integration time is increased.
(4) Extract the dynamic response time history of the structure.

## 3. Benchmark Finite Element Model

### 3.1. Case Study

The bridge studied is an asymmetric single-tower cable-stayed bridge with a single cable plane, which has been in operation for more than 20 years. The bridge is located in Fuzhou City, Fujian Province, China. The bridge deck is a two-way six-lane urban trunk road. An image of the bridge is shown in Figure 1. The main span of the bridge is 238 m, and the side span is divided into three spans of 76 + 56 + 47 m by two auxiliary piers. Stay cables are fan-shaped and arranged in the central separation zone of the main girder. The bridge is equipped with basin rubber bearings at piers #1, #3, #4 and #5, and the overall arrangement is shown in Figure 2. The main girder has a single box and three rooms section with a section height of 3.3 m. The bridge tower has a tower-beam-pier consolidation system, and the transverse direction of the bridge is an inverted Y-shape. Except for the solid tower column below the bridge deck, it has a single-chamber box section. The main girder and pylon are made of C50 concrete, and its standard value of compressive strength is 32.4 MPa. The whole bridge is designed with 58 pairs of stay cables. The cables are made of prefabricated parallel steel wires, and the standard value of the tensile strength is 1570 MPa. The numbers of the stay cables are #M1 . . . #M29 and #S1 . . . #S29, where M stands for the main span and S stands for the side span.

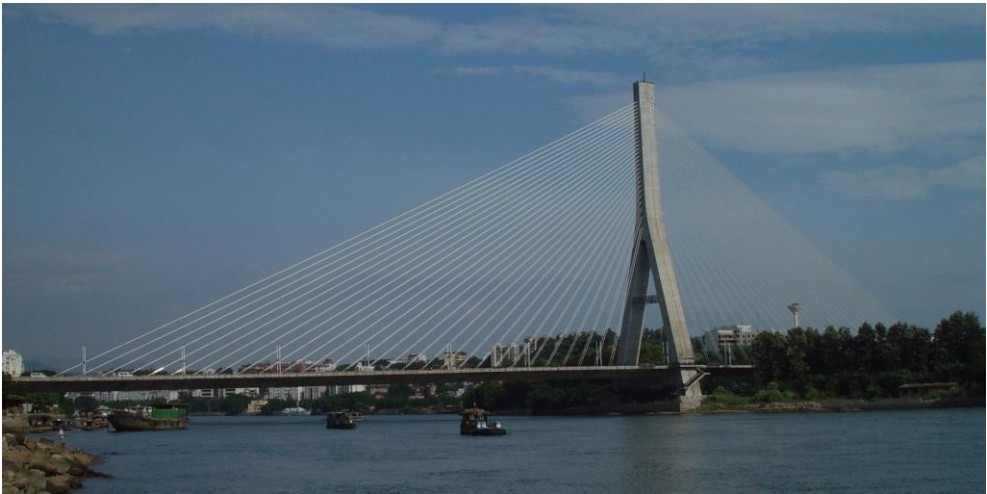

**Figure 1.** Image of the bridge.

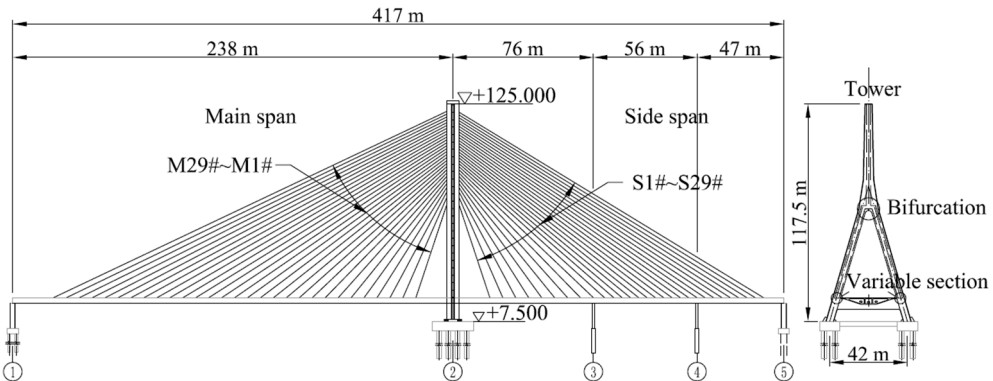

**Figure 2.** Layout of the single tower cable-stayed bridge.

### 3.2. Finite Element Model

The 3D finite element model of the bridge was established using ANSYS. The pylon, main girder and pier were all simulated using Beam188 element, which has the ability to withstand tension, compression, torsion and bending. The cable was simulated using Link10, which has the characteristics of stress stiffening, large deformation, and element life and death. In order to anchor the cables to the main girder and pylon, a Beam4 element with sufficient stiffness was used to simulate the anchorage connection of cables. Mass21 with large deformation and life-and-death characteristics was used to simulate the dead load of the structure. The cable tower was directly connected with the main girder in a rigid way, which restricted all the degrees of freedom of the tower bottom node and the pier bottom node, thus ensuring the accuracy of the simulation of each component unit. The finite element model contained 2159 elements and 1083 nodes, as shown in Figure 3.

In order to obtain a model that reflected the actual working state of the cable-stayed bridge, the cable control module of MIDAS was adopted, and the measured cable force was brought in to solve the dead load equilibrium state of the finite element model until the calculated and measured values of the cable force and line shape met the accuracy requirements. The comparison between the calculated values of the finite element model and the measured values of the bridge is shown in Figure 4. Among them, the measured cable force was obtained using the vibration frequency method, and the linear shape of the main girder was measured using the linear monitoring points arranged in the early stage. Due to the limitation of space, only the upstream side cables are given.

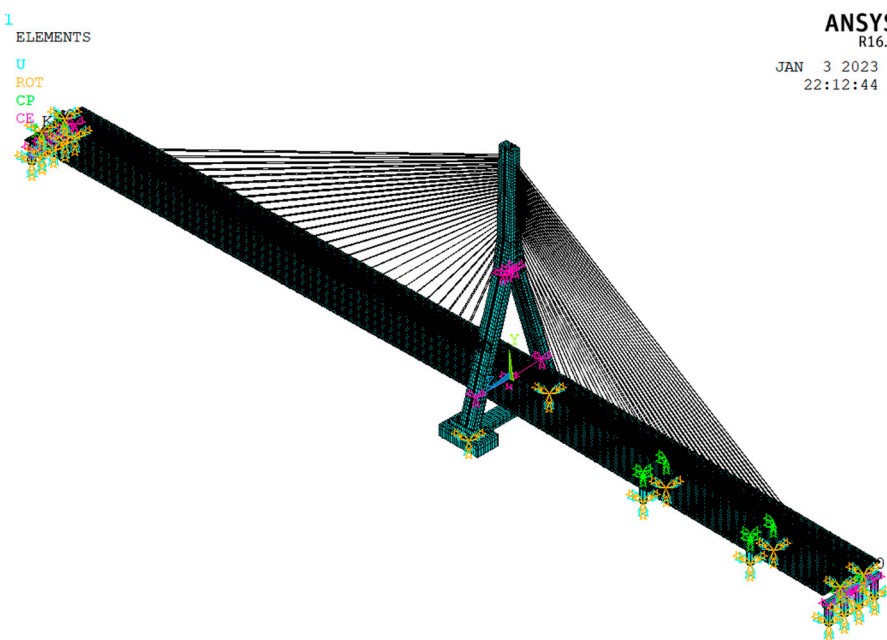

**Figure 3.** Finite element model of the bridge.

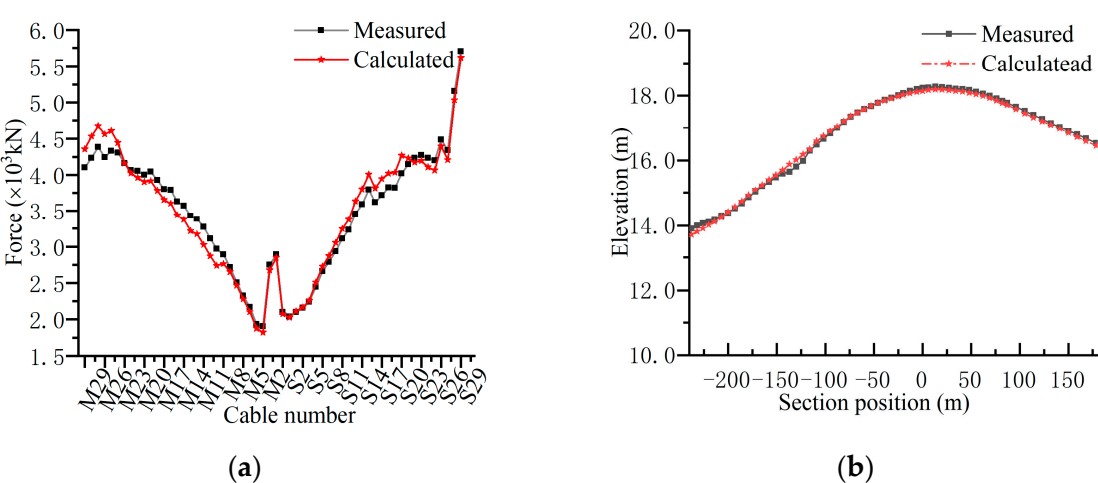

(**a**)  (**b**)

**Figure 4.** Comparison between the calculated value and measured value: (**a**) cable force comparison; (**b**) comparison of main girder alignment.

As can be seen from Figure 4, the calculated cable force of the model was basically consistent with the measured value, and the calculated elevation of the main girder was in good agreement with the measured elevation. The finite element model could truly reflect the stress state of an example single-tower cable-stayed bridge and could be used for the dynamic study of cable breakage.

## 4. Cable-Breaking Simulation

To investigate the sudden breaking of the cables of a single-tower cable-stayed bridge, the semi-dynamic simulation method was adopted in this study. Considering the non-linearity of the structure, the Newmark method was used for the dynamic calculation. To obtain the maximum dynamic response of the structure, the most unfavorable time step was determined to be 0.001 s after many trial calculations. Because the influence of damping on the structure could not be ignored, the Rayleigh damping formula was used to determine the damping matrix of the structure. Referring to the Code for Wind-Resistant Design Specification for Highway Bridges [22], the damping ratio of the structure was set

to 0.02. Because the cable breakage mainly affected the vertical bending of the main girder and the vertical bending of the cable tower, the damping coefficient was calculated by selecting the structural mode frequencies of the first-order vertical bending of the girder and the first-order vertical bending of the cable tower.

### 4.1. Cable-Breaking Time

The research shows that the shorter the duration of cable breaking, the more intense the structural vibration and the greater the peak value of the structural dynamic response [20]. Through the trial calculation of the breaking duration of various cables, it was determined that the most unfavorable situation was 0.02 s, which was about 1/100th of the first-order vibration period of the cable-stayed bridge.

The breakage of the cables caused the structure to vibrate violently, and the stress of the cables increased sharply in an instant, which easily led to the breakage of damaged adjacent cables one after another. Due to the continuous change in stress during the vibration, the interval between broken cables that led to the maximum vibration response exists and was not easily perceived. The most unfavorable interval time of cable breaking was obtained using dichotomy iterations, and the solution steps are shown in Figure 5.

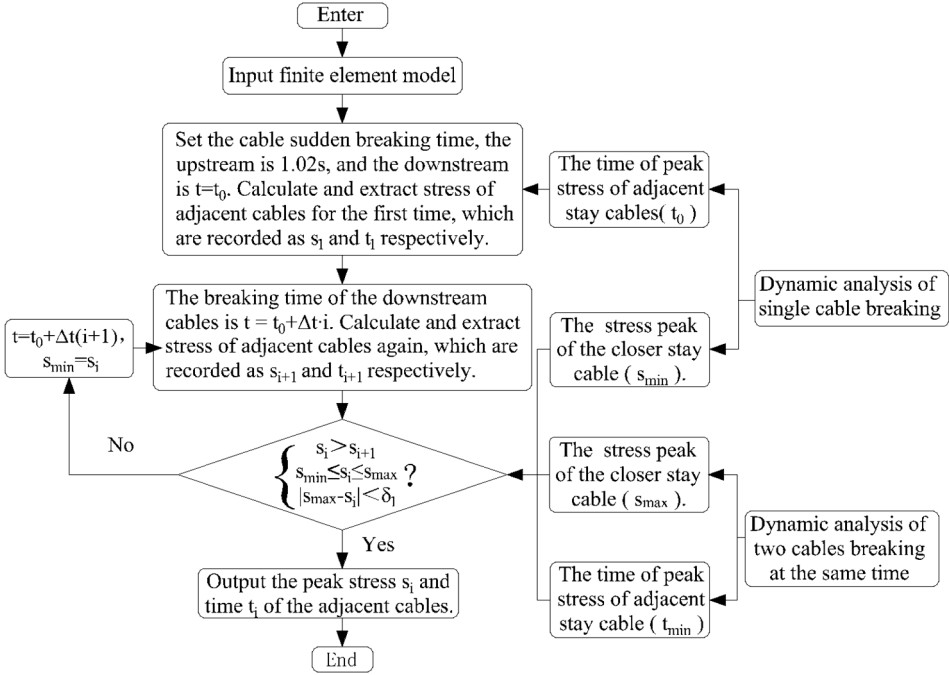

**Figure 5.** Solution diagram of the cable-breaking interval time.

### 4.2. Analysis Condition Selection

Because the stress levels of different cables of a cable-stayed bridge are different, and they are in the natural environment for a long time, the erosion and damage of cables are also uncertain, and thus, the possible breaking positions are random. Therefore, when the cables in different positions are broken, the influence on the structural stress and deformation is also different. Thus, the dead load deformation of the structure after the cables were broken at different positions was calculated, and the maximum deflections of the main girder and the displacement of the tower top were extracted, as shown in Figure 6.

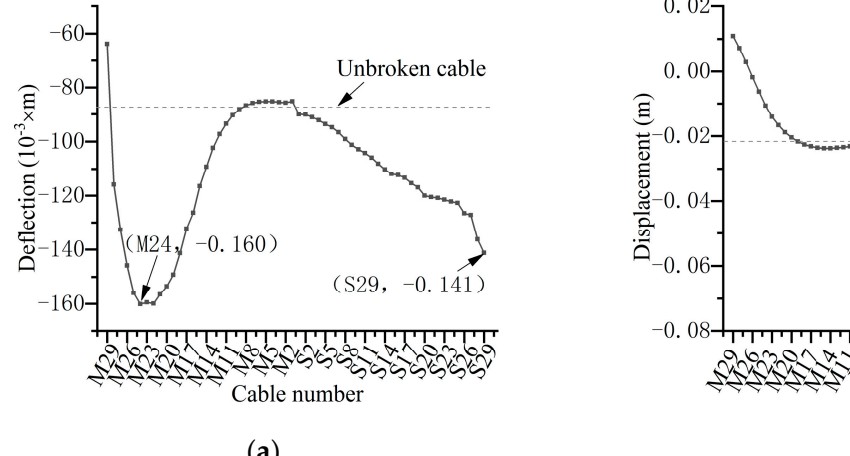

(**a**)　　　　　　　　　　　　　　　　　　　　　　　　　(**b**)

**Figure 6.** Maximum displacement of the structure after the cables were broken at different positions: (**a**) maximum deflection of the main girder; (**b**) displacement of the cable tower at the top. Note: Negative values in the figure indicate that the main girder was deflected downward and the pylon was biased toward the main span.

It can be seen from Figure 6a that the maximum deflection of the main girder first decreased and then increased with the breaking of cables at different positions from #M29~#M1. After the #S1~#S29 cables were broken, the maximum deflection of the main girder shows a nonlinear decreasing trend. The breakage of the #M24 cable in the main span had the greatest influence on the deflection of the main girder, and the maximum deflection of the main girder was −0.160 m, with an increase of 0.073 m. The breakage of the #S29 stay cable on the side span also had a great effect on the deflection of the main girder, and the maximum deflection of the main girder was −0.141 m, with an increase of 0.054 m. As shown in Figure 6b, with the breakage of the #M29~#S29 cables, the displacement at the top of the cable tower was less nonlinear. The breakage of the #S29 cable had the greatest influence on the cable tower alignment, and the displacement increase was 0.051 m. As far as the structural deformation of the cable-stayed bridge is concerned, the breakage of the #M24 cable in the main span and the #S29 cable in the side span had the greatest influence on the structural deformation; therefore, these two types of cables were selected for the subsequent dynamic analysis of successive breakage.

## 5. Results

In order to reveal the influence of cable breakage on structural stress and deformation, this study analyzed the dynamic response of this cable-stayed bridge structure under constant load and cable breakage impact. The cable-breaking mode was upstream cable breaking → downstream cable breaking. The breaking objects were the #M24 and #S29 cables. The breaking times of adjacent cables were 1.812 s and 1.982 s, respectively, according to the dichotomy iterative solution process proposed in Section 4.1. To explain the dynamic response results, the observation points of the structure and ten adjacent cables were selected as the analysis objects. The observation points of the main girder and cable tower are shown in Figure 7.

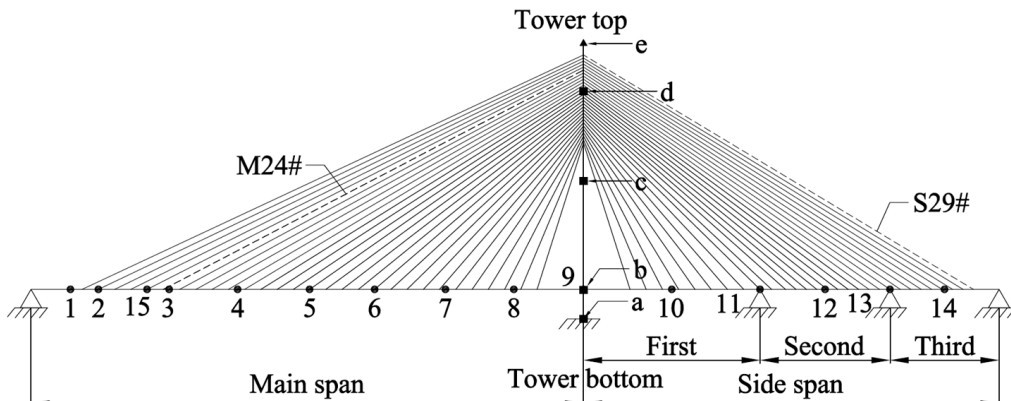

**Figure 7.** Observation point position. Note: Observation points #1 and #15 were the most unfavorable position and the maximum displacement position of the main beam under dead load, respectively. Points #2~#8 were the main span bisector, and observation points #9~#14 were the cable tower, the middle span of each side span and the pier. The points b~d were the bifurcation, variable cross-section and the most unfavorable position of the straight tower, respectively.

*5.1. Cable Stress*

The dynamic response of adjacent cables was extracted, and only the stress time history of adjacent cables is given due to the limitation of space, as shown in Figure 8. The initial stress $S_0$, peak stress $S_{dyn}$ and quasi-static stress $S_s$ of the cable were compared, and the stress increase and dynamic amplification factor ($DAF_1$) of the cable were calculated, as shown in Figure 9.

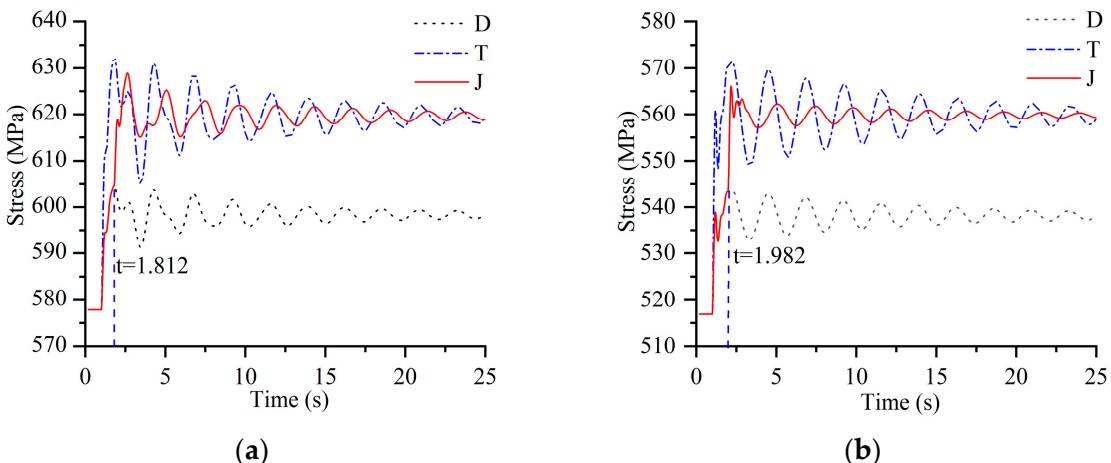

**(a)** **(b)**

**Figure 8.** Stress response of the nearest cables: (**a**) #M23 cable stress after #M24 broke; (**b**) #S28 cable stress after #S29 broke. Note: D indicates a single cable breaking, T indicates that pairs of cables were broken at the same time, and J indicates that pairs of cables were broken one after another.

It can be seen from Figure 8 that the fluctuation amplitude of the cable stress after successive breaking was smaller than that of simultaneous breaking, but it was closer to the stress of simultaneous breaking. The stress of adjacent cables increased sharply due to the breaking of upstream cables, and reached the stress peak at 1.812 s and 1.982 s, respectively, and then increased sharply again due to the breaking of downstream cables, which shows that the breaking time point that caused the maximum vibration response of the structure could be obtained using dichotomy iterations, and the structural vibration caused by the continuous breaking of cables was not as severe as the simultaneous breaking.

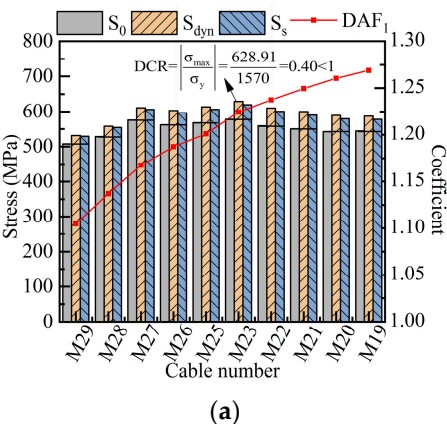 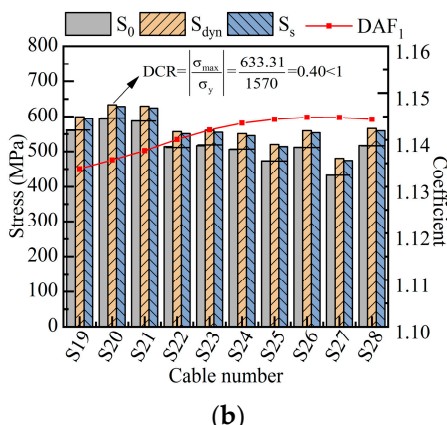

(**a**)  (**b**)

**Figure 9.** Stress of ten adjacent cables: (**a**) stress of adjacent cables after #M24 broke; (**b**) stress of adjacent cables after #S29 broke. Note: $S_{dyn} - S_0$ is the peak increase and $S_s - S_0$ is the quasi-static increase. The stress response values can be found in Table A1 in Appendix A.

As shown in Figure 9, the closer to the broken cable position, the greater the stress increase of the cable, which shows obvious proximity. The DCR of the cable stress was less than 1.0, which shows that the residual cable strength met the dynamic stress requirement of cables breaking one after another. As can be seen from Figure 9a, the peak stress of #M23 cable was the largest, and the peak stress was 628.91 MPa. The peak increase and quasi-static increase were 51.04 MPa and 41.68 MPa, respectively, and the ratio between them was 1.224. With the number of stay cables decreasing, the $DAF_1$ gradually increased, both of which were within 1.10~1.27, and the $DAF_1$ of the #M19 cable was larger. It can be seen from Figure 9b that the peak stress of the #S20 cable was the largest, and its peak increase was 40.05 MPa, which was 1.135 times that of the quasi-static stress increase. The stress increase of the #S28 cable was the largest, and its peak increase was 48.96 MPa, which was 1.144 times that of the quasi-static stress increase. With the increase in the number of stay cables, the $DAF_1$ gradually increased, ranging from 1.13 to 1.15, and the $DAF_1$ values of the #S26 and #S27 stay cables were larger.

### 5.2. Stress of the Main Beam

The stress dynamic response of the observation points of the main girder after the #M24 and #S29 cable breaks were extracted. The initial stress $S_0$, the maximum dynamic stress $S_{dyn}$ during vibration and the quasi-static stress $S_s$ were compared, the dynamic influence coefficient (DIC) and dynamic amplification coefficient ($DAF_2$) of the cable breaks were calculated, and the demand capacity ratio (DCR) of the main girder stress was calculated using Formula (5), as shown in Figures 10 and 11.

From Figure 10, it can be seen that the stress of the main girder increased greatly near the broken cable, and the stress in the span of side 3 changed greatly because the deflection of the main girder near the broken cable increased at the moment when the cable was broken. Although the broken cable caused the stress near the broken cable and the corresponding position on the opposite side of the cable tower to increase sharply, the stress fluctuations in other positions were small. The increase in dynamic stress of the #M24 cable breaking was greater than that of the #S29 cable breaking, and the DCR of the main beam stress was less than 1.0, which shows that the strength of the main beam could meet the dynamic stress demand of cable breaking.

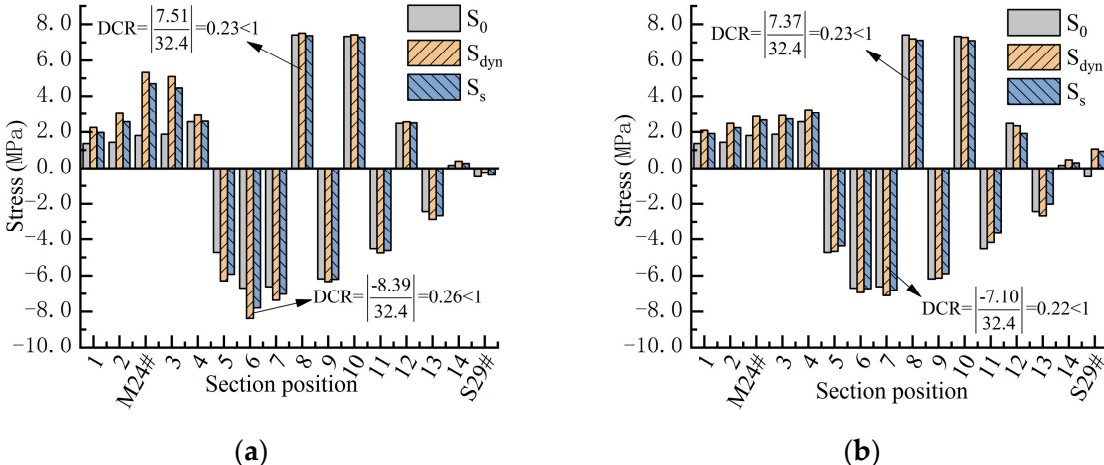

**Figure 10.** Stress of the main beam and DCR: (**a**) girder stress after #M24 broke; (**b**) girder stress after #S29 broke. The stress response of the main beam can be found in Table A2 in Appendix A.

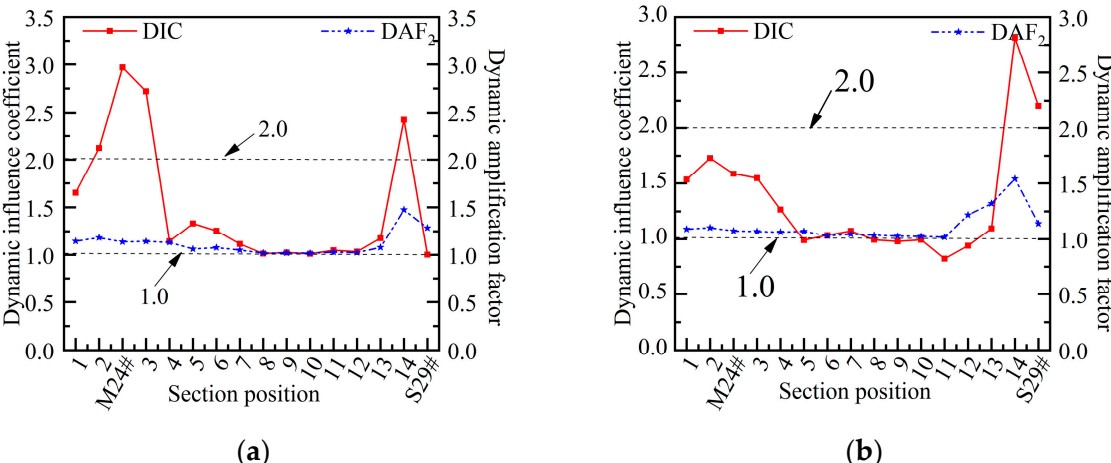

**Figure 11.** DIC and DAF$_2$ of main beam stress: (**a**) stress coefficient after #M24 broke; (**b**) stress coefficient after #S29 broke. The specific values can be found in Table A2 in Appendix A.

As shown in Figure 11a, DIC varied from 1.01 to 2.968, and the position of the broken cable is the largest. DAF$_2$ varied from 1.02 to 1.48, and it was the largest in the middle span of side 3. From Figure 11b, it can be found that the DIC of the main span fluctuated between 0.82 and 2.88, which was close to 1.0 except at the adjacent position, which also shows that the stress of the main span increased and the stress of the different span decreased at the moment when the #S29 cable broke. The DAF$_2$ of the main beam stress varied from 1.02 to 1.58, and the side 3 span was the largest.

### 5.3. Tower Stress

Three kinds of compressive stresses ($S_0$, $S_{dyn}$, $S_s$) at the designated observation points of the pylon after the cables successively broken were compared, and DIC, DAF$_2$ and DCR of the pylon stresses were calculated, as shown in Table 1.

As shown in Table 1, the DIC of the cable tower stress was between 0.955 and 1.4, and DAF$_2$ was between 1.0 and 1.1. The demand capacity ratio of the cable tower stress was less than 1.0, which shows that the strength of the cable tower could meet the dynamic stress demand of the #M24 and #S29 cables breaking successively. Although the breakage of the #M24 cable caused stress fluctuation in the tower, the stress variation amplitude was small, and the quasi-static stress was less than the initial stress. The maximum dynamic stress of the tower bottom was 8.29 MPa, which was 1.063 times that of the quasi-static stress, and

the maximum increase in the dynamic stress was 4.5%. As the fracture of the #S29 cable caused the stress fluctuation of the pylon, the stress variation amplitude was small, and the quasi-static stress was less than the initial stress. In the most unfavorable part of the cable tower, the maximum dynamic stress increased by 32.9%, and the maximum dynamic stress was 1.036 times that of the quasi-static stress.

**Table 1.** The stress comparison of the cable tower after the cables were successively broken.

| Broken Object | Observation Point | Stress (MPa) | | | DIC | DAF$_2$ | DCR |
|---|---|---|---|---|---|---|---|
| | | $S_0$ | $S_{dyn}$ | $S_s$ | | | |
| #M24 | a | −7.93 | −8.29 | −7.80 | 1.045 | 1.063 | 0.26 |
| | b | −13.40 | −13.55 | −13.15 | 1.011 | 1.030 | 0.42 |
| | c | −6.94 | −7.02 | −6.70 | 1.012 | 1.048 | 0.22 |
| | d | −7.59 | −7.25 | −6.98 | 0.955 | 1.039 | 0.22 |
| #S29 | a | −7.93 | −8.94 | −8.28 | 1.127 | 1.081 | 0.28 |
| | b | −13.40 | −14.43 | −13.94 | 1.077 | 1.035 | 0.45 |
| | c | −6.94 | −8.10 | −7.75 | 1.167 | 1.044 | 0.25 |
| | d | −7.59 | −10.09 | −9.73 | 1.329 | 1.036 | 0.31 |

### 5.4. Structural Displacement

The deflection of observation point #15 of the main girder and the response value of horizontal displacement at the top of the cable tower were extracted, and the DIC and DAF$_2$ of the displacement after the cable breaking were calculated, as shown in Figure 12.

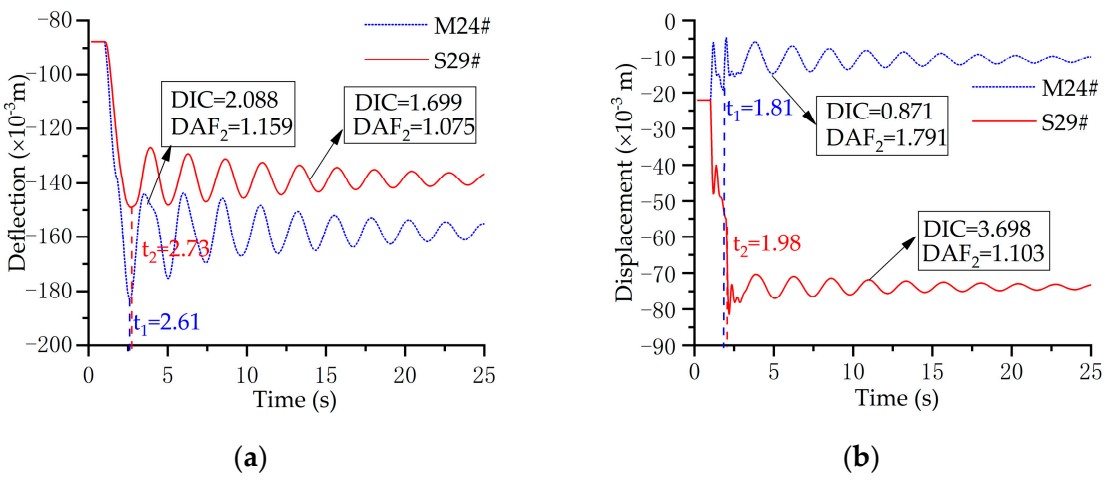

**Figure 12.** Displacement time history of cables breaking in succession: (**a**) deflection of #15 observation point of main girder; (**b**) displacement of the top of the cable tower.

As shown in Figure 12a, the dynamic deflection of the #S29 cable was smaller than that of the #M24 cable. The maximum dynamic deflection at the observation point of the main girder #15 was −0.183 m, which was less than the L/500 stiffness limit of 0.476 m required by the Chinese design standard [23]. The DIC of the maximum deflection of the main girder was 2.088, and DAF$_2$ was 1.159. It can be seen from Figure 12b that the top displacement of the tower changed sharply due to the breakage of the upstream cables, and changed sharply again due to the breakage of the downstream cables. The breakage of the #M24 cable reduced the displacement of the tower top, the maximum dynamic displacement was −0.019 m, and the dynamic amplification factor of displacement was 1.791. The breakage of the #S29 cable increased the displacement, and the maximum dynamic displacement was −0.081 m, which was 3.689 times that of the initial displacement, and the dynamic amplification factor of displacement was 1.103.

## 6. Conclusions

In this study, the cable that had the greatest influence on structural deformation was selected as the breaking object, and the dynamic influence coefficient (DIC), dynamic amplification coefficient (DAF) and demand capacity ratio (DCR) were taken as the breaking analysis indexes (3D indexes). The influence of the cable breaking on the dynamic response of asymmetric single-tower cable-stayed bridge was studied using the semi-dynamic simulation method, and the conclusions were as follows:

(1) A method was proposed to solve the time point of continuous breaking of cables, which could obtain the time of continuous breaking that caused the maximum vibration response of the structure. The peak response value of cables breaking in succession was less than that of the cables breaking at the same time, and the structural vibration caused by the cables breaking in succession was not as severe as that caused by the cables breaking at the same time.

(2) The strength of the remaining cables could meet the dynamic stress requirement of instantaneous cable breakage, and the cable with the largest dynamic stress increase was the adjacent cable. The closer to the broken cable, the greater the dynamic stress increase. With the change in cable number, the DAF increased nonlinearly and changed within 1.10~1.30.

(3) Although the broken cable caused the stress near the broken cable and the corresponding position on the opposite side of the cable tower to increase sharply, the stress fluctuations in other positions were small. The strength of the main girder met the dynamic stress requirement of the broken cable vibration, and the maximum dynamic displacement of the main girder was less than the stiffness limit. The dynamic stress increase of the #M24 cable breaking was greater than that of the #S29 cable breaking; the dynamic stress increase was the largest at the broken cable position; and the stress DAF was between 1.01 and 2.52, with the largest at the mid-span position of side 3.

(4) In the structural vibration caused by successive breakdowns, the strength of the cable tower could meet the demand of dynamic stress. The stress change was relatively small, but the displacement of the tower top changed greatly. The breaking of the longest cable in the side span had the greatest influence on the stress of the tower, with a maximum increase in the dynamic stress of 36.3% and the maximum dynamic displacement of the tower top exceeding 3.0 times that of the initial displacement.

**Author Contributions:** Conceptualization, Y.C. and H.C.; formal analysis, Y.C. and S.W.; methodology, J.Z. and H.C.; software, Y.C. and S.W.; resources, K.H. and H.C.; writing—review and editing, Y.C. and S.W.; supervision, S.W., H.C. and Y.C. All authors have read and agreed to the published version of the manuscript.

**Funding:** This research was funded by the projects of "scientific research on intelligent management of long-span double-deck steel truss combined beam and bridge life cycle" grant number [00501917] and "research on real-time monitoring system of important bridges" grant number [0050229916]. And The APC was funded by [0050229916].

**Institutional Review Board Statement:** We choose to exclude this statement because the study did not require ethical approval.

**Informed Consent Statement:** Informed consent was obtained from all subjects involved in the study.

**Data Availability Statement:** We choose to exclude this statement because the study did not report any data.

**Acknowledgments:** The authors acknowledge the support of China Railway Bridge Research Institute Co., Ltd. and Fuzhou Municipal Engineering Center.

**Conflicts of Interest:** The authors declare no conflict of interest.

## Appendix A

**Table A1.** The stress responses of 10 adjacent cables after the cables were successively broken.

| Broken Object | Cable Number | Stress (MPa) | | | $S_{dyn} - S_0$ | $S_s - S_0$ | DAF$_1$ | DCR |
|---|---|---|---|---|---|---|---|---|
| | | $S_0$ | $S_{dyn}$ | $S_s$ | | | | |
| #M24 | #M29 | 507.78 | 531.26 | 529.04 | 23.48 | 21.26 | 1.105 | 0.34 |
| | #M28 | 528.48 | 557.78 | 554.25 | 29.30 | 25.76 | 1.137 | 0.36 |
| | #M27 | 575.66 | 610.95 | 605.89 | 35.30 | 30.23 | 1.167 | 0.39 |
| | #M26 | 562.01 | 602.94 | 596.49 | 40.93 | 34.47 | 1.187 | 0.38 |
| | #M25 | 567.70 | 613.56 | 605.89 | 45.86 | 38.19 | 1.201 | 0.39 |
| | #M23 | 577.87 | 628.91 | 619.56 | 51.04 | 41.68 | 1.224 | 0.40 |
| | #M22 | 559.20 | 610.05 | 600.29 | 50.85 | 41.09 | 1.237 | 0.39 |
| | #M21 | 550.45 | 599.72 | 589.88 | 49.27 | 39.43 | 1.250 | 0.38 |
| | #M20 | 542.37 | 588.90 | 579.28 | 46.53 | 36.92 | 1.260 | 0.38 |
| | #M19 | 544.13 | 586.97 | 577.89 | 42.84 | 33.76 | 1.269 | 0.37 |
| #S29 | #S19 | 560.33 | 598.84 | 594.30 | 38.51 | 33.96 | 1.134 | 0.38 |
| | #S20 | 593.28 | 633.31 | 628.53 | 40.03 | 35.25 | 1.136 | 0.40 |
| | #S21 | 587.72 | 629.26 | 624.23 | 41.54 | 36.51 | 1.138 | 0.40 |
| | #S22 | 513.88 | 556.74 | 551.47 | 42.85 | 37.59 | 1.140 | 0.35 |
| | #S23 | 516.63 | 560.75 | 555.26 | 44.11 | 38.62 | 1.142 | 0.36 |
| | #S24 | 505.98 | 551.29 | 545.60 | 45.31 | 39.62 | 1.144 | 0.35 |
| | #S25 | 473.51 | 519.93 | 514.08 | 46.42 | 40.57 | 1.144 | 0.33 |
| | #S26 | 512.23 | 559.67 | 553.67 | 47.44 | 41.44 | 1.145 | 0.36 |
| | #S27 | 432.28 | 480.58 | 474.47 | 48.30 | 42.19 | 1.145 | 0.31 |
| | #S28 | 516.92 | 565.88 | 559.70 | 48.96 | 42.78 | 1.144 | 0.36 |

Note: $S_{dyn} - S_0$ is the peak increase and $S_s - S_0$ is the quasi-static increase.

**Table A2.** The stress response of the main beam after the cables were successively broken.

| Broken Object | Section Position | Stress (MPa) | | | DIC | DAF$_2$ | DCR |
|---|---|---|---|---|---|---|---|
| | | $S_0$ | $S_{dyn}$ | $S_s$ | | | |
| #M24 | #1 | 1.36 | 2.25 | 1.96 | 1.653 | 1.144 | 0.07 |
| | #2 | 1.42 | 3.02 | 2.56 | 2.123 | 1.182 | 0.09 |
| | M24# | 1.80 | 5.33 | 4.69 | 2.968 | 1.137 | 0.16 |
| | #3 | 1.87 | 5.10 | 4.46 | 2.721 | 1.142 | 0.16 |
| | #4 | 2.55 | 2.92 | 2.58 | 1.144 | 1.131 | 0.09 |
| | #5 | −4.72 | −6.28 | −5.92 | 1.331 | 1.060 | 0.19 |
| | #6 | −6.68 | −8.39 | −7.79 | 1.255 | 1.076 | 0.26 |
| | #7 | −6.60 | −7.36 | −7.01 | 1.116 | 1.050 | 0.23 |
| | #8 | 7.41 | 7.51 | 7.37 | 1.012 | 1.018 | 0.23 |
| | #9 | −6.18 | −6.32 | −6.19 | 1.023 | 1.021 | 0.20 |
| | #10 | 7.33 | 7.42 | 7.29 | 1.012 | 1.017 | 0.23 |
| | #11 | −4.52 | −4.74 | −4.61 | 1.049 | 1.028 | 0.15 |
| | #12 | 2.47 | 2.55 | 2.49 | 1.033 | 1.023 | 0.08 |
| | #13 | −2.43 | −2.86 | −2.65 | 1.176 | 1.078 | 0.09 |
| | #14 | 0.16 | 0.39 | 0.27 | 2.420 | 1.473 | 0.01 |
| | #S29 | −0.478 | −0.479 | −0.37 | 1.002 | 1.285 | 0.01 |
| #S29 | #1 | 1.36 | 2.08 | 1.91 | 1.534 | 1.090 | 0.06 |
| | #2 | 1.42 | 2.46 | 2.23 | 1.728 | 1.103 | 0.08 |
| | #M24 | 1.80 | 2.85 | 2.65 | 1.585 | 1.074 | 0.09 |
| | #3 | 1.87 | 2.90 | 2.71 | 1.546 | 1.069 | 0.09 |
| | #4 | 2.55 | 3.23 | 3.04 | 1.266 | 1.063 | 0.10 |
| | #5 | −4.72 | −4.67 | −4.36 | 0.990 | 1.072 | 0.14 |
| | #6 | −6.68 | −6.92 | −6.71 | 1.036 | 1.031 | 0.21 |
| | #7 | −6.60 | −7.10 | −6.80 | 1.075 | 1.044 | 0.22 |
| | #8 | 7.41 | 7.37 | 7.12 | 0.994 | 1.036 | 0.23 |
| | #9 | −6.18 | −6.05 | −5.89 | 0.979 | 1.027 | 0.19 |
| | #10 | 7.33 | 7.28 | 7.10 | 0.994 | 1.026 | 0.22 |
| | #11 | −4.52 | −3.72 | −3.65 | 0.822 | 1.018 | 0.11 |
| | #12 | 2.47 | 2.32 | 1.91 | 0.940 | 1.217 | 0.07 |
| | #13 | −2.43 | −2.66 | −2.02 | 1.097 | 1.319 | 0.08 |
| | #14 | 0.16 | 0.46 | 0.30 | 2.812 | 1.539 | 0.01 |
| | #S29 | −0.48 | 1.05 | 0.92 | 2.203 | 1.140 | 0.03 |

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
