# Peer review of "Research on Dynamic Response of a Single-Tower Cable-Stayed Bridge with Successive Cable Breaks Based on a 3D Index"

_applsci, doi:10.3390/app13169197_

Round 1
Reviewer 1 Report
This reviewer recommends publication of the manuscript for the fact it presents practical engineering results backed by simulación.
The manuscript needs English language review.
Author Response
The manuscript needs English language review.
Response: The author has checked the full-text English writing, and has made some corrections to the problems with grammatical errors.
Reviewer 2 Report
A comprehensive numerical analysis of a 20-year-old single-tower cable-stayed bridge was carried out in the work.Using the ANSYS program, the change in stresses and displacements caused by breaking one or two bridge cables was determined.
The work does not contain any new elements.
It is only a parametric study of the impact of a hypothetical cable failure on the mechanical response of the structure.
Although the work can hardly be considered original, it can be published after completing the description of the results.
The work analyzes a real single-tower cable-stayed bridge. The obtained results should be the basis for formulating general conclusions regarding this type of construction.
For example: should similar calculations be made at the design stage of a similar bridge, and if so, to what extent?
In addition: it should be checked how many broken cables will lead to a catastrophe.
The work analyzes the situation when the only load is the weight of the bridge and the impact of broken cables. However, a bridge failure never occurs in isolated conditions. The work lacks any commentary on the impact of external loads, e.g. wind. It should be verified whether in such cases breaking even a single cable will lead to an immediate failure of the bridge.
There is also no comment on the impact of the seismic load.Without appropriate supplements, the work should not be published.
The authors should also consider performing additional calculations, such as taking into account the wind load and checking the possibility of failure of the structure weakened by a broken cable.
Reviewer 3 Report
Regardless that this article lacks of experimental validation in laboratory, the numerical work is carried out correctly and the results obtained are interesting and are in accordance with results obtained previously in similar woks performed by different researchers. This article provides new results for an asymmetric cable-stayed bridge and reinforces several of the conclusions obtained previously in diverse studies. The topic studied is very important for the construction industry, which helps to avoid bridges’ collapses with the corresponding economic losses and human fatalities. In general, the article is well written and I recommend this article to be published after a minor revision:
* The Introduction is somewhat short; the authors should include more works/references about similar works.
* When using the equals sign (=), a blank space must be included before and after the sign, but that sometimes does not happen. Likewise, a blank space between a parameter value and its measurement unit must be included. Please, re-check the entire article.
* The title of Section 3.1 must start with the first letter in upper case: “Case” instead of “case”.
* In caption of Figure 4, red color must not be used.
* The title of Section 5 should be changed to “Results” instead of “Result”.
* In the title of Section 5.2, the word “Main” should be written in lower case (following the format of the other titles/subtitles).
* Be sure that all the variables mentioned in Equations and Figures/Tables are described (explanation of the meaning) and the format is uniform (do not use lower case for a variable and then upper case for the same variable, italics/normal, bold/normal, different letter fonts/sizes, etc.), since there are several inconsistencies.
* Schematic diagrams with the followed processes to simulate/analyze the different scenarios of cables’ breakage should be provided for a quick understanding (Section 2).
* In Figure 1, units must be included.
* The name and location of the real-life bridge considered for performing the FEM model/simulations should be mentioned in Section 3.1, as well as a picture should be provided.
* After line 183, a brief description of how the measured values of the bridge were obtained should be provided, as well as some pictures, if possible.
* The quality of Figure 4 should be increased.
* In Figure 5, the legend for X axis of Figure 5a is “cable number” and for Figure 5b is just “cable” and they must be same.
* Figure 7 is confused (plots and captions) in the sense of which cables were removed and where the responses were measured. Authors should re-check this situation and make a detailed description about the cables removed for each case and the points of measuring.
* Why in caption of Figure 7a “M23” is mentioned and into the plot “M24”?
* In Figure 8b, the largest peak increment and largest quasi-static increment are supposed to occur in cable 28, if that does not happen please justify.
* The results obtained in Figures 8-10 should be provided in Tables so that the readers can analyze the results quickly. In those tables, percentage variations should be included and the most important results, which are useful to obtain conclusions, should be highlighted.
* In Figure 10a, if the broken cable is M24, why the maximum value of DAF2 corresponds with a position close to S29?
* In line 305, the authors say that in Figure 10a DIC varies from 1.01 to 3.75; however, the maximum value of DIC is around 3.0.
* In line 307, the authors mentioned Figure 9b but the Figure analyzed is 10b.
* In line 308, the value “0.99” seems to be wrong. The correct one is around 0.80.
Minor editing of English language required.
Author Response
Minor editing of English language required.
Response :The author has checked the full-text English writing, and has made some corrections to the problems with grammatical errors.
Please see the attachment

Reviewer 4 Report
The title is unclear - 3D index?
The dynamic analysis method is not specified. Referring to the naming from the ANSYS system is not enough. No modal parameters of the structure and no damping parameters are given. Has the FEM model been validated based on measurements of the real structure (shapes and eigenfrequencies, damping)? Where did the forces measured in the stayes come from?
If the M24 and S29 stays were broken, the description in Fig. 7 is unclear.
Why doesn't the displacement history in Fig. 11 start at 0.00?
Round 2
Reviewer 4 Report
It is a pity that the authors did not provide the parameters of the dynamic analysis. Time step, method of analysis (pure Newnark?). Validation of the FEM model, for example by comparing the form and frequency of vibrations with reality, is not provided. Damping can also be identified in this way. The assumed damping ratio of 0.02 seems to be large for a cable-stayed bridge. In conclusion, it should be stated that there are many aspects that may affect the course and results of the analyses. Nevertheless, the work is worth publishing.
Author Response
Point 1: It is a pity that the authors did not provide the parameters of the dynamic analysis. Time step, method of analysis (pure Newnark?).
Response 1: The semi-dynamic simulation method was adopted in this paper. Considering the nonlinearity of cable-stayed bridge structure, Newmark method was used for dynamic analysis. To obtain the maximum dynamic response of the structure, the most unfavorable time step is determined to be 0.001 s after several trial calculations.
Point 2: Validation of the FEM model, for example by comparing the form and frequency of vibrations with reality, is not provided. Damping can also be identified in this way. The assumed damping ratio of 0.02 seems to be large for a cable-stayed bridge.
Response 2: The finite element model is verified by comparing the calculated values of the model with the measured values of the bridge, including cable force and main girder alignment, as shown in Figure 4. Due to the lack of dynamic load test data of the bridge, the author cannot obtain the measured frequency data. Through many trial calculations and referring to the research of many scholars, the most unfavorable situation is determined to be 0.02.